# TopoSD: Topology-Enhanced Lane Segment Perception with SDMap Prior

## Abstract

Recent advances in autonomous driving systems have shifted towards reducing reliance on high-definition maps (HDMaps) due to the huge costs of annotation and maintenance. Instead, researchers are focusing on online vectorized HDMap construction using on-board sensors. However, sensor-only approaches still face challenges in long-range perception due to the restricted views imposed by the mounting angles of onboard cameras, just as human drivers also rely on bird's-eye-view navigation maps for a comprehensive understanding of road structures. To address these issues, we propose to train the perception model to "see" standard definition maps (SDMaps). We encode SDMap elements into neural spatial map representations and instance tokens, and then incorporate such complementary features as prior information to improve the bird's eye view (BEV) feature for lane geometry and topology decoding. Based on the lane segment representation framework, the model simultaneously predicts lanes, centrelines and their topology. To further enhance the ability of geometry prediction and topology reasoning, we also use a topology-guided decoder to refine the predictions by exploiting the mutual relationships between topological and geometric features. We perform extensive experiments on OpenLane-V2 datasets to validate the proposed method. The results show that our model outperforms state-of-the-art methods by a large margin, with gains of +6.7 and +9.1 on the mAP and topology metrics. Our analysis also reveals that models trained with SDMap noise augmentation exhibit enhanced robustness.

## 1 Introduction

Autonomous driving has witnessed remarkable advancements in recent years, becoming increasingly integral to the future of transportation. As a crucial component, perceiving the complex road scenarios to estimate the lane geometry and road topological connections is critical not only to the downstream planning, but also to ensuring the reliability and explainability of the overall system. As the foundational infrastructure for autonomous driving, high-definition maps (HDMaps) can provide a detailed and accurate source of road structures and geometries. Nevertheless, the annotation and maintenance costs of HDMaps are substantial, which poses limitations on their scalability across widespread areas. To alleviate these issues, recent researches such as (Li et al., 2022a; Liu et al., 2023; Liao et al., 2022; 2023b; Ding et al., 2023) are exploring how to construct online HD maps using onboard sensor input powered by deep learning models. However, relying solely on onboard sensors to accurately recognize lane-level geometry and topology remains challenging in real-world environments. They may produce low-quality lane lines or erroneous topology connections due to constrained camera views and limited visual ranges, and the situations are particularly exacerbated during severe weather conditions or occlusion.

It is natural that human drivers maneuver vehicles not only by observing the surroundings of the vehicle, but also by referring to navigation maps, i.e. standard-definition maps (SDMaps), or a memory map in one's mind. SDMaps encompass the road structures, typically consisting of road networks, intersections, and other basic geographic features. With the localization of the global position system (GPS), the corresponding SDMaps serve as a quick visual prompt of the surrounding real environment and complement the sensor input. Importantly, compared to HDMaps, SDMaps are easier to obtain and are updated more frequently to accommodate new road changes, which makes SDmaps preferable prior information for model input to complement the pure sensor input.

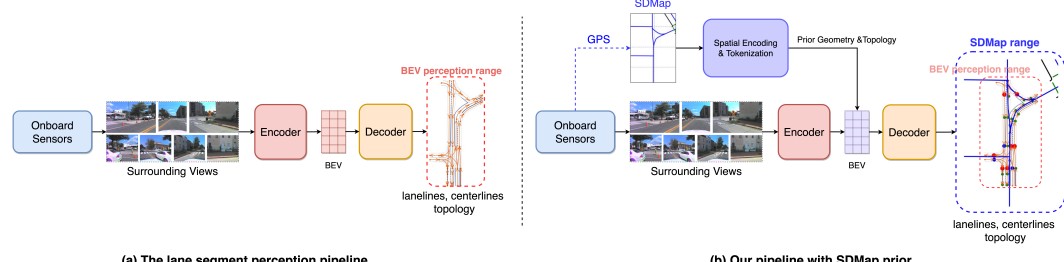

Figure 1: Comparison between the previous lane segment perception pipeline and ours. We incorporate SDMap information as prior to enhance the geometry and topology learning.

As shown in Figure 1, based on the framework of predicting lane segments only using cameras, we incorporate the SDMap into the perception model to solve the problem of understanding driving scenes and improving the map construction on lane geometry and topology.

To make use of SDMaps, we encode the elements in the map into a representation that the neural network can learn from. We employ two distinct encoding methodologies: 1) spatial map encoding – drawing various SDMap attributes into 2D spatial maps aligned with the BEV range; 2) map tokenization – encoding SDmap information, e.g. the class and coordinates of SDMap polylines in a larger range into token vectors. Consequently, the key information of road networks is encoded into such representations and is then fused into the BEV feature to help online map construction and improve prediction confidence. From the intrinsic characteristics of these encoding methods, the advantage of the former is that the geometry and topology information of the connected road polylines can be encoded in the spatial distribution of SD neural maps, thereby complementing the BEV feature. Conversely, the latter method offers the advantage of encoding a broader range of road information beyond the BEV perception range and capturing the global topology relationships between different SDMap instances.

While the goal of the lane segment perception task is to unify the geometry and topology modeling, the mutual promotion between geometrical and topological features remains unexplored. The common practice for topology prediction typically involves using independent branches for geometry and topology prediction tasks. To further exploit reciprocal benefits between two prediction tasks, we design a topology-guided decoder operating recursively to gradually encourage prediction consistency among queries using an adjacent matrix conveying the topology information. It takes into account both the successor and predecessor of a lane, which further improves the accuracy of both topological and geometrical predictions. We conduct extensive experiments on the lane segment perception benchmark OpenLaneV2 dataset. Compared to current state-of-the-art methods, our model demonstrates substantial improvements, achieving a +6.7 increase in mAP, a +9.1 increase in the topology metric, and a +5.5 increase in the OLS score.

Our contributions can be summarized as follows: (1) We incorporate the SDMap as prior information to tackle the task of lane segment perception. We propose two complementary SDMap encoding methods to leverage the topology and geometry information of SDMap to enhance the BEV perception. (2) We propose a Topology-Guided Decoder to exploit the mutual promotive relationships between geometrical and topological features, enhancing the predictions of both geometry and topology. (3) We conduct extensive experiments on the OpenLaneV2 benchmark. The results show that our model outperforms the counterpart methods with a large margin, and achieves state-of-the-art performance in the lane segment perception task. We also conduct a comprehensive analysis of the impact of SDMap errors or noise on performance and propose potential strategies to enhance model robustness.

## 2 RELATED WORK

**Lane detection** & **Online HD Map construction** Lane detection is the common task of detecting lane elements in the road scenes. Many of them focus on single-view lane recognition and they can be classified into segmentation-based (Pan et al., 2018; Abualsaud et al., 2021), anchor-based (Tabelini et al., 2021; Qin et al., 2020; Xiao et al., 2023) and keypoint-based (Ko et al., 2021; Wang et al.,

2022) methods. Recently, great progress has been made in the field of online HD map construction. BEV-LaneDet (Wang et al., 2023c) and HDMapNet (Li et al., 2022a) adopt a typical rasterized map representations, which outputs segmentation results and embeddings for clustering. However, such methods need extra post-processing to generate maps for downstream planning modules. In contrast, vectorized map representation induces an end-to-end learning paradigm, with various methods (Liu et al., 2023; Liao et al., 2022; 2023b; Ding et al., 2023; Zhang et al., 2024; Qiao et al., 2023a;b) having been proposed. MapTR (Liao et al., 2022) and MapTRv2 (Liao et al., 2023b) employ point-level queries for each lane instance and an end-to-end learning paradigm, effectively enhancing the perceptual accuracy of vectorized maps. PivotNet (Ding et al., 2023) proposes a compact pivot-based map representation and attempts to model the topology in dynamic point sequences by introducing the concept of sequence matching. GeMap (Zhang et al., 2023) proposes to learn Euclidean shapes and relations of map instances beyond basic perception. MapTracker (Chen et al., 2024) formulates the map construction as a tracking task, uses the memory buffer to ensure consistent reconstructions over time and augments the mAP metrics with consistency checks.

**Lane Topology Reasoning.** Lane topology reasoning is directly related to the detection of centerlines and their connectivity. STSU (Can et al., 2021) introduces a DETR-like network to detect centerlines, and uses a MLP to infer their connectivity to form a directed graph. CenterLineDet (Xu et al., 2023) regards centerlines as vertices in a graph and employs a model trained through imitation learning to update the topology. TopoMLP (Wu et al., 2023) uses two high-performance detectors and two MLP networks for lane detection and topology reasoning. LaneGAP (Liao et al., 2023a) uses a path-wise approach to translate the lane graph into continuous and complete paths and a heuristic-based algorithm to recover the lane graph. TopoNet (Li et al., 2023) explicitly models the connectivity of centerlines and integrates traffic elements to learn a comprehensive understanding of the driving scene. LaneSegNet (Li et al., 2024) introduces a new representation of lane segments. It leverages both geometric and topological modeling, further enhancing the prediction ability of road structure. In this paper, we use the same representation of lane segment but introduce the SDMap information as prior and design a topology-guided decoder to further improve the accuracy of geometry and topology predictions.

**Map Fusion**. Recent approaches make attempts to leverage some prior map for online HD mapping. Neural Map Prior (Xiong et al., 2023) builds a neural representation of global maps as a strong prior map, which are fused and updated when conducting local map inference. (Gao et al., 2023) proposes using satellite maps to complement onboard sensors to improve HD map construction. The satellite image features are fused into the BEV feature using a hierarchical fusion module. StreamMapNet (Yuan et al., 2024) fuses the temporal information from the memory feature updated by history frames to improve performance. MapEX (Sun et al., 2023) proposes to improve online HD construction using existing maps. It encodes the elements of HDMap into the map queries and leverage the decoder to utilize the existing map. There are some concurrent works incorperate SDMap as extra inforamtion to improve the onlien HD mapping. P-MapNet (Jiang et al., 2024) incorporates both SDMap and HDMap as prior to improve the model performance. It uses attention-based architecture to fuse the relevant SDMap skeletons for map construction and pre-trains a HDMap prior module to refine the map segmentation results. SMERF (Luo et al., 2023) integrates SD maps into online map construction. It encodes the class and coordinates of SDMap polylines into vectors using a Transformer encoder, and the map features are fused into the BEV feature using cross-attentions. The proposed map tokenization in this paper builds upon this method to encode a larger range of SDMap. Additionally, we propose a spatial representation encoding to enhance the geometric and topological attributes of SDMap.

## 3 METHOD

We aim to tackle the task of driving scene structure perception and reasoning, particularly focusing on the lane detection and the topology prediction. Built upon the lane segment based representation (Wang et al., 2023b; Li et al., 2024), we exploit standard-definition maps (SDMaps) as prior to enrich the perception information in BEV, as SDMaps can offer rough road geometry and topology information to generate map structure. To exploit the mutual relationships between the topological and geometrical feature, we employ a Topology-Guided Decoder (TGD) equipped with a topology-guided self-attention mechanism to optimize the centerline geometry and topology using a predicted adjacent matrix.

Figure 2: The overall model architecture. The model receives perspective images from cameras arranged in a surrounding view configuration and a locally aligned SDMap as inputs. The images are processed by the image backbone to obtain multi-scale image features. The polylines of SDMap are encoded as two representations – a 2D-shaped SD feature map and a set of vectorized SD tokens. We adopt a BEVFormer-like encoder to extract BEV features. The SD feature map is added to the BEV queries and BEV features. The SD tokens interact with BEV queries via cross-attention. Then we use a Topology-Guided Decoder to predict the lane segment results. SA denotes the Self-Attention layer.

### 3.1 LANE SEGMENT PERCEPTION TASK

In this task, a lane segment is a minimum unit to predict which contains a centerline, a left-boundary, and a right-boundary of a lane instance in form of polylines, denoted as $\mathbb{V} = \{\boldsymbol{v}_c, \boldsymbol{v}_l, \boldsymbol{v}_r\}$ respectively. For the left or the right boundary, the line type of them $\{a_l, a_r\}$ are defined within: non-visible, solid, and dashed. Besides, following LaneSegNet (Li et al., 2024), we convert the pedestrian crossing into the format of lane segment and exclude the prediction of road boundary.

The task of lane segment perception is not only to accurately detect the geometries of lane segment but is to generate the topological relationships between detected lane segments, i.e., the lane graph. This lane graph is represented as a directed graph $\mathcal{G} = (V, E)$. Each lane segment $\mathbb{V}$ is denoted as a node in the set $V$, and the edges in set $E$ represent connections between lane segments. Each edge signifies a directed connection between two lane segments that have preceding and succeeding relationships.

### 3.2 SDMAP ENCODING AND FUSION

We use the SDMaps as the extra input, which conveys information about the road type, the road shape and topological connection. To make full use of them, we encode the map entities in the map into a representation that the neural network can learn from by using two distinct encoding methodologies as follows.

**1) spatial map encoding** . This encoding method is to draw SDMap elements into 2D canvas maps. Assuming a 2D canvas is drawn according to the geometry and types of roads in the SDMap, the pixels in this canvas convey the SDMap information locally and the road structure in the bird's eye view can be expressed in the 2D maps. In light of this, the SDMap polyline elements are first encoded into different canvas maps. These maps are drawn with thick lines to describe the geometry, connections, shape, and types of the roads. And we employ cosines and sines of the inclination angle of the road line segments to express the curvature of the roads. And then these maps are processed by a CNN to achieve the SD feature $\mathbf{F}_s \in \mathbb{R}^{d \times h \times w}$. Refer to Appendix A for more details on encoding.

**2) map tokenization**. As the SD features only encode the SDMap information locally, we use another approach – map tokenization to encode the class and coordinate information in a global scope. Inspired by the polyline sequence representation in SMERF (Luo et al., 2023), we encode $S$ polyline instances in SDMap as $S$ token vectors, each of which is combined by a one-hot category vector representation with $K$ dimensions and $N$ point coordinate embeddings with $c$ dimension. In other words, the dimension of each SD token vector is $N \cdot c + K$. The $S$ SD token vectors are then sent to

a Transformer encoder to model the internal relationships among these SD elements and transformed into the $D$ dimension token-based map feature $\mathbf{T}_s \in \mathbb{R}^{S \times D}$.

**SDMap Pre-fusion.** Given this new input modal, how and where to incorporate such SDMap information is critical to the model performance. Considering that SDMap only contains coarse road structure information, we advocate for introducing the SDMap to the model at an earlier stage of processing, rather than integrating it during the final lane prediction phase when the local information is much more crucial.

To this end, we propose to pre-fuse SDMap in the stage of constructing BEV feature and expect to reduce the possible negative interference when the inconsistency between sensor data and map occurs. We adopt a BEVFormer-based (Li et al., 2022b) encoder to generate the BEV feature. We add SD features $\mathbf{F}_s$ to initial BEV queries $B_q \in \mathbb{R}^{D \times h \times w}$ and the output of the BEVformer Encoder. In the stage of BEV feature learning, the BEV queries can further aggregate image features from surrounding perspective-view images via a cross-attention mechanism. After this cross-attention, another cross-attention layer is appended to query the BEV feature with SD tokens $\mathbf{T}_s$. The purpose of this design is to enable bev queries to select the most relevant tokens to fuse. Finally, the obtained SD-enhanced BEV feature $\mathbf{F}_B \in \mathbb{R}^{D \times h \times w}$ are sent to the decoder for further processing. In experiments, we find all these designs are indispensable for achieving good performance.

### 3.3 TOPOLOGY-GUIDED DECODER

Although this task of lane segment perception is to uniformly learn the geometry and topology of the road structure, the mutual influence of topology and geometry has not been fully explored in current approaches. In LaneSegNet (Li et al., 2024), the topology information is inferred using the final queries after the geometrical locations of centerlines have been predicted. However, this approach ignores the fact that topology information may affect the geometric position of the centerline. Intuitively, for two lanes with topological connections, their geometric endpoints are also connected with each other. If carefully designed, an approach should benefit from the relationship between lane topology and geometric layout. Therefore, we insert a topology-guided self-attention mechanism in each decoder layer, which allows the predicted topology information to influence the prediction of geometric information layer by layer, thus promoting mutual interaction between topology information and geometric positions.

We employ a deformable DETR (Zhu et al., 2020) style decoder to map the SDMap-enhanced BEV feature to final outputs through multiple heads. The learnable instance queries $Q \in \mathbb{R}^{N \times D}$ represent lane segments. For the interactions with the BEV feature, we still keep the Lane Attention mechanism proposed in LaneSegNet to cross-attention with BEV feature $\mathbf{F}_B$, obtaining its outputted instance queries.

**Topology-guided Self Attenion Mechanism**. After the Lane Attention, we insert Topology-guided Self-Attenion. In Topology-guided Self-Attenion, a topology head is used to predict the topology adjacency matrix between lane segments $\boldsymbol{M}_{topo} \in \mathbb{R}^{N \times N}$. Then we use this predicted topology matrix to fuse the geometrical information of the predecessor and the successor. More specifically, we leverage the adjacency matrix $\boldsymbol{M}_{topo}$ to represent the topological connectivity. Each element in the matrix has a value between 0 and 1, a higher score representing a higher connectivity possibility. An element in the matrix $\boldsymbol{M}_{topo}$ indexed with $(i, j)$ represents the possibility of the endpoint of $i$-th lane segment connected with the start point of $j$-th lane segment. Assuming the feature outputted by the self-attention is $\mathbf{F}$. By left-multiplying $\mathbf{F}$ with $\boldsymbol{M}_{topo}$, we obtain the successor connection enhanced feature: $\boldsymbol{F}_{succ} = \boldsymbol{M}_{topo}\boldsymbol{F} \in \mathbb{R}^{N \times D}$ Similarly, left-multiplying the transpose of $\boldsymbol{M}_{topo}$ with $\boldsymbol{F}$ yields the predecessor connection enhanced feature $\boldsymbol{F}_{prede} = \boldsymbol{M}_{topo}^T\boldsymbol{F}$. We carry out these two operations right after the self-attention layer in the decoder. These three features $\boldsymbol{F}$, $\boldsymbol{F}_{succ}$, and $\boldsymbol{F}_{prede}$ are concatenated with MLPs to form the final topology-enhanced feature:

$$\mathcal{F} = MLP(\text{Concatenate}(\boldsymbol{F}, MLP(\boldsymbol{F}_{succ}), MLP(\boldsymbol{F}_{prede})) \in \mathbb{R}^{N \times D}. \quad (1)$$

As a result, these enhanced features $\mathcal{F}$ have incorporated the original self-attention information with successor and predecessor connection features, providing a more comprehensive representation of the interactions between different instances. We embed this topology-guided attention operation in each decoder layer. Through multiple decoder layers, the geometric information of lanes can be optimized by the topology matrix and the topology adjacent matrix in each decoder layer is predicted

Table 1: Comparison with State-of-the-Art method on the lane segment perception task. We mainly compare the proposed method with the official results of LaneSegNet on subset_A set. The $\text{TOP}_{lsls}$ is based on the newly updated metric. The results of TopoNet and MapTR are from the paper (Li et al., 2024). For P-MapNet, we follow the official implementation regarding the cross-attention, OSM-CNN and the downsampling settings. We downsample the BEV feature and SD feature by 4 times (cross-attention with size of $50 \times 25$) for their cross-attentions and then recover their sizes.

| Method | SDMap Encoder | Epoch | mAP | $\text{AP}_{ls}$ | $\text{AP}_{ped}$ | $\text{TOP}_{lsls}$ |
|---|---|---|---|---|---|---|
| TopoNet (Li et al., 2023) | - | 24 | 23.0 | 23.9 | 22.0 | - |
| MapTRv2 (Liao et al., 2023b) | - | 24 | 28.5 | 26.6 | 30.4 | - |
| LaneSegNet (Li et al., 2024) | - | 24 | 33.5 | 32.0 | 34.9 | 25.4 |
| LaneSegNet + SMERF (Luo et al., 2023) | Transformer | 24 | 37.1 | 37.2 | 36.9 | 30.5 |
| LaneSegNet + P-MapNet (SD cross-attn.) (Jiang et al., 2024) | OSM CNN | 24 | 30.0 | 29.2 | 30.8 | 25.1 |
| LaneSegNet + P-MapNet (SD cross-attn.) (Jiang et al., 2024) | ResNet-18 | 24 | 33.2 | 32.6 | 33.9 | 28.3 |
| **Ours-1 (LaneSegNet + Spatial Enc. + Tokenization)** | ResNet-18+Transformer | 24 | 39.9 (+6.4) | 37.8 (+5.8) | 41.9 (+7.0) | 32.0 (+6.6) |
| **Ours-2 (LaneSegNet + Spatial Enc. + Tokenization + TGD)** | ResNet-18+Transformer | 24 | **40.2** (+6.7) | **38.6** (+6.6) | **41.7** (+6.8) | **34.5** (+9.1) |

by the updated lane segment queries, thereby mutually enhancing the accuracy of both topology and geometric predictions.

**Heads**. Like LaneSegNet, we adopt multiple MLP heads to decode the class, line types, centerline coordinates, and offsets from each instance query. The left and right boundary lines can be obtained by subtracting and adding the predicted offset to the predicted centerline, respectively: $\hat{\boldsymbol{v}}_l = \hat{\boldsymbol{v}}_c - \hat{\boldsymbol{o}}$, $\hat{\boldsymbol{v}}_r = \hat{\boldsymbol{v}}_c + \hat{\boldsymbol{o}}$. And the final output instance queries are sent to the topology head to predict the adjacency matrix. Due to the fact that the centerlines of lane segments are connected by start points and end points. Therefore, we design a connection head to predict the adjacency matrix using start points and end points information. In this connection head, each query is firstly transformed into the end embedding $\boldsymbol{E}_e \in \mathbb{R}^{N \times D_e}$, and start embedding $\boldsymbol{E}_s \in \mathbb{R}^{N \times D_e}$ by two distinct MLPs. We use the inner product as an association score, and hence the adjacency matrix is computed as $\boldsymbol{M}_{topo} = \boldsymbol{E}_e \boldsymbol{E}_s^T \in \mathbb{R}^{N \times N}$.

# 4 EXPERIMENTS

**Dataset.** We conducted experimental validation on the subset A set of OpenLaneV2 Dataset (Wang et al., 2023a). OpenLaneV2 is a large-scale 3D lane dataset and comprises 1000 segments of various scenarios, including daytime, nighttime, sunny, rainy, urban, rural, and more. Each scenario lasts approximately 15 seconds, effectively providing feedback on the algorithm's efficacy. The annotations of lane segments and the perception range are within $\pm 50m$ along the x-axis and $\pm 25m$ along the y-axis. For the used SDMaps, we pre-process the original SDMap polylines to a large range within $\pm 100m$ along the x-axis and $\pm 50m$ along the y-axis, the center of which is still aligned with the center of the perception range.

**Metric**. As we mainly focus on the lane segmentation perception task, we report the results on the specifically designed metrics based on the lane segment distance $\mathcal{D}_{ls}$, following (Li et al., 2024). It induces the average precision $\text{AP}_{ls}$ and $\text{AP}_{ped}$ to evaluate the accuracy of lane segments, pedestrian crossings and the mean AP is computed as the average of $\text{AP}_{ls}$ and $\text{AP}_{ped}$. We use $\text{TOP}_{lsls}$ to evaluate the accuracy of topological connections between centerlines. See more information about the implementation details and metrics in Appendix C.

## 4.1 COMPARISON WITH STATE-OF-THE-ART

Due to that the LaneSegNet is the first method performing on the lane segment perception benchmark, we mainly compare our models with it on the overall metrics and report the results of other HDMap constructing methods. As shown in Table 1, ours-1 model with SDMap pre-fusion substantially outperforms the LaneSegNet with +6.4 on the mAP and +6.6 on the $\text{TOP}_{lsls}$ metric. Such results demonstrate that the SDMap can provide a strong prior to help generate the maps and improve the predictions on lane segments' geometry and topology. Further enhanced by the Topology-Guided Decoder (TGD), our model achieves a new set of state-of-the-art performance with 40.2% on mAP and 34.5% on $\text{TOP}_{lsls}$, gaining obvious improvements with **+6.7** on mAP and **+9.1** on $\text{TOP}_{lsls}$ compared with LaneSegNet. To ensure a fair comparison with contemporary works, SMERF (Luo

et al., 2023) and P-MapNet (Jiang et al., 2024), we integrated them with LaneSegNet. For the LaneSegNet model incorporating P-MapNet, we utilized our spatial encoded maps as SDMap inputs. The comparative results presented in Table 1 demonstrate that our proposed models (such as 'our-1') exhibit superior performance across multiple metrics.

To show the overall performances on the complex road scene perception and understanding, we train our model the map bucket with multiple tasks on OpenLaneV2 based on the lane segment representation. The pedestrian and road boundary are detected by an additional MapTR head (Liao et al., 2022). The traffic elements are detected by a Deformable DETR head (Zhu et al., 2020). The hyper-parameters are roughly set. As shown in Table 2, our model still surpasses the LaneSegNet model on all metrics.

Table 2: Comparison with State-of-the-Art method on OpenLaneV2 map element bucket. We mainly compare the proposed method with LaneSegNet by running its official bucket configuration.

| Method | Epoch | $DET_{ls}$ | $DET_a$ | $DET_t$ | $TOP_{lsls}$ | $TOP_{lste}$ | OLS score |
|---|---|---|---|---|---|---|---|
| LaneSegNet | 24 | 27.4 | 18.4 | 38.0 | 24.1 | 20.9 | 35.7 |
| Ours | 24 | 37.0 | 21.6 | 40.4 | 33.6 | 24.0 | 41.2 |

## 4.2 ABLATION STUDY

In this section, we conduct ablations to validate the proposed SDMap encoding and fusion methods, as well as the Topology-guided decoder.

**Ablations on SD encoding**. Since we propose two types of SDMap encodings, we validate the effectiveness of two encoding methods respectively, as well as the effect of combining both SD encoding methods. As shown in Table 3, our findings indicate that both encoding methods independently bring significant gains, and their combination results in even higher gains. This means that two types of SD encoding methods can play different roles without conflicts at different levels, particularly for the map tokenization method that encodes a larger range of SDMap road polylines than the spatial map encoding.

**Ablations on the fusion method**. For the SD map tokenization, we use cross-attention layers in the BEV Encoder to fuse SD tokens with the BEV feature by default. However, especially for the utilization of SD spatial map encodings, there are still multiple choices to fuse the SD feature.

We observe that fusing the SD feature into the BEV query (Exp-5) results in greater improvements in mAP (+4.8) and $TOP_{lsls}$ (+6.4) compared to fusing the SD feature (Exp-4) into the BEV feature, which showed improvements in mAP (+3.6) and $TOP_{lsls}$ (+5.1). Such results imply that incorporating 2D spatial SD structure information in the BEV query may provide a stronger prior and give more room for the BEV query to aggregate online visual information from cameras. We find that adding SD feature to both the BEV query and BEV feature still gains further improvements (Exp-6 in Table 3).

**Ablation on Topology-guided decoder**. Based on the SD fusion model, we validate the effectiveness of the topology-guided decoder. The results in Table 3 show that the topology-guided decoder can gain improvements of 0.8 and 2.5 on the $AP_{ls}$ and $TOP_{lsls}$ metrics, which means that the geometry and topology lane segments are specifically optimized thanks to the topology enhanced decoder.

Table 3: Ablations on SDMap encoding and fusion methods, as well as the Topology-Guided Decoder. The column of Fusion Position only indicates where to fuse the SD feature when using spatial encoding.

| Exp | Spatial Enc. | Tokenization | Fusion Position | Decoder | mAP | $AP_{ls}$ | $AP_{ped}$ | $TOP_{lsls}$ |
|---|---|---|---|---|---|---|---|---|
| 1 | - | - | - | LaneSeg | 33.5 | 32.0 | 34.9 | 25.4 |
| 2 | ✓ | - | BEV feat. | LaneSeg | 36.8 | 34.6 | 39.1 | 28.9 |
| 3 | - | ✓ | - | LaneSeg | 37.2 | 36.9 | 36.9 | 30.5 |
| 4 | ✓ | ✓ | BEV feat. | LaneSeg | 39.1 | 37.3 | 40.9 | 30.7 |
| 5 | ✓ | ✓ | BEV query | LaneSeg | 38.3 | 37.2 | 39.4 | 31.8 |
| 6 | ✓ | ✓ | BEV feat. + BEV query | LaneSeg | 39.9 | 37.8 | 41.9 | 32.0 |
| 7 | ✓ | ✓ | BEV feat. + BEV query | Topo-Guided | 40.2 | 38.6 | 41.7 | 34.5 |

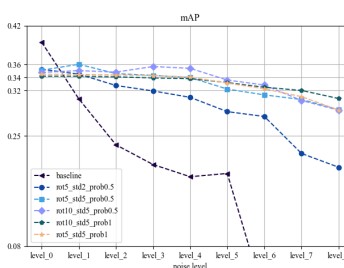

(a) The mAP metric under different SDMap noise level.

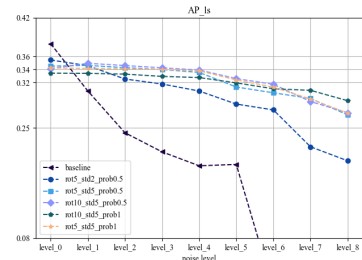
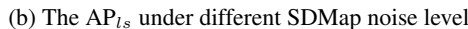

(b) The $AP_{ls}$ under different SDMap noise level.

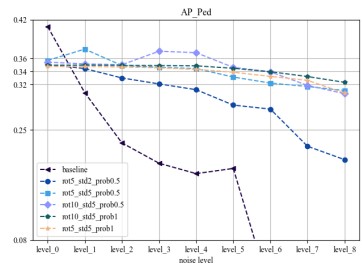

(c) The $AP_{ped}$ under different SDMap noise level.

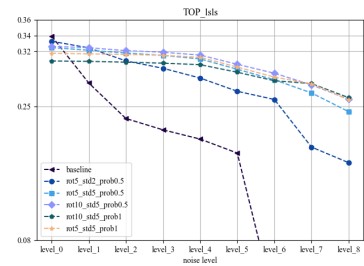

(d) The $TOP_{lsls}$ under different SDMap noise level.

Figure 3: The model performances under different levels of SDMap noise. Each curve represents the same model trained under some condition of adding SDMap noise.

## 4.3 STUDY ON THE ERROR PROBLEMS OF SDMAP

In practical applications, SDMap errors are a crucial consideration, particularly concerning system localization and map annotation. These errors can arise from factors such as imprecise GPS signals and ambiguous road centerline positions in the forward direction. To simulate these errors in real-world scenarios, we conducted experiments involving the addition of random shifting and rotational noise during training and testing.

Table 4: Performances on different settings of SDMap random noise.

| Method | Training SDMap noise | Testing SDMap noise | mAP | $TOP_{lsls}$ |
|---|---|---|---|---|
| LaneSegNet | - | - | 33.5 | 25.4 |
| Baseline SD model (Ours-2) | - | - | 40.2 | 34.5 |
| Baseline SD model (Ours-2) | - | rot5_std5_prob0.5 | 23.6 (-41.3%) | 24.4 (-29.3%) |
| Noisy SD model (Ours-2) | rot5_std5_prob0.5 | - | 35.1 | 32.5 |
| Noisy SD model (Ours-2) | rot5_std5_prob0.5 | rot5_std5_prob0.5 | 34.6 (-1.4%) | 31.8 (-2.2%) |

Assuming the *baseline* SD model is trained using the original SDMap annotations, we train the same model with different SDMap noise injection by adding a random shifting sampled from a Gaussian distribution and a random rotation sampled from a uniform distribution. We set three variables: the standard deviation (*std*, with meter as its unit) of the Gaussian distribution for shifting noise and the maximum rotation angle (*rot*) for the random rotation, and the probability (*prob*) of whether to add random noise. We control these variables to combine several configurations such as *rot5_std2_prob0.5*.

In Table 4, we present comparative results demonstrating the model performance when trained and tested with or without adding random SDMap noise. In Figure 3, we train the baseline model using different noise configurations and evaluate their performance across noise levels ranging from level-0 to level-8 (see details in Appendix B). Our findings indicate that when testing without adding noise to the SDMap input, the baseline model outperforms the models trained with noisy SDMap input. This suggests that the model heavily relies on the geometric information provided by the SDMap for accurate predictions. However, as the level of noise increases, the performance of the baseline model gradually deteriorates, eventually collapsing at the highest noise level. Interestingly, the models trained with noisy SDMaps, despite experiencing performance degradation, demonstrate relatively

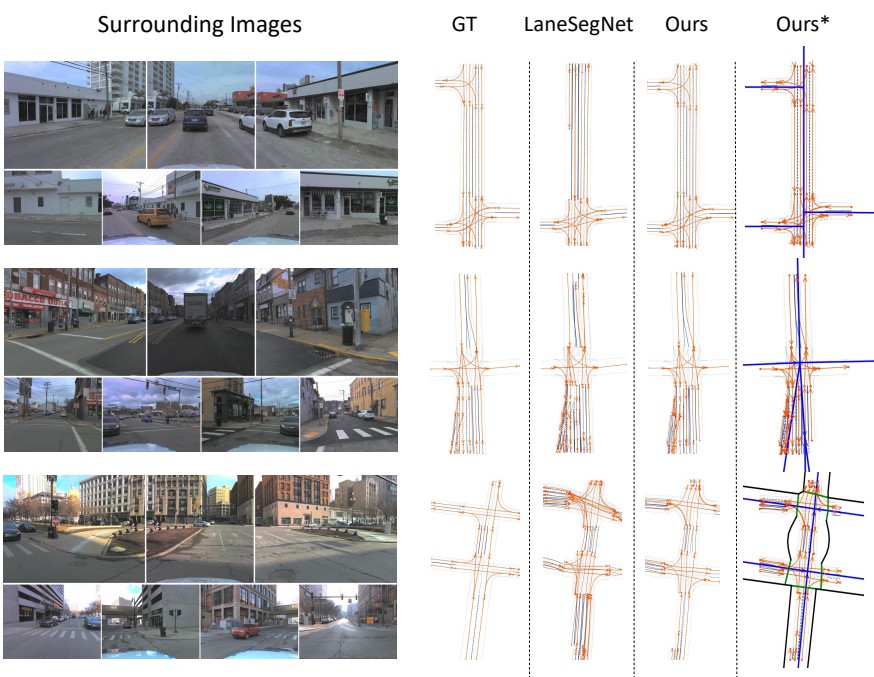

Figure 4: Visualization results on some cases. We compare our model with the ground truth, and the prediction results of LaneSegNet. * means the predicted lane segments with the input SDMap polylines. The blue, black and green lines represent roads, sidewalks and crosswalks in SDMap.

better performance at higher noise levels. This phenomenon implies that these models develop a robust reliance on both the SDMap and visual features, enabling them to perform well even in the presence of SDMap errors or shifting.

## 4.4 COMPUTATIONAL COMPLEXITY AND EFFICIENCY ANALYSIS

Table 5: Computational efficiency analysis. The BEV feat. size and SD feat. size both indicate the resolutions when fusing them together. The inference speeds are tested on a single Tesla V100-32G GPU with a batch size of 1. $N_{sd}$ represents the maximum number of SDMap elements in a batch.

| Method | SDMap Encoder | Inference Speed (FPS) | Model Params. | BEV feat. size | SD feat. size |
|---|---|---|---|---|---|
| LaneSegNet | - | 4.3 | 45.4M | 200×100 | - |
| LaneSegNet + SMERF | Transformer | 4.0 | 48.6M | 200×100 | $N_{sd}$ |
| LaneSegNet + P-MapNet (SDMap cross-attention) | Small OSM CNN | 3.9 | 51.4M | 50 ×25 | 50×25 |
| LaneSegNet + P-MapNet (SDMap cross-attention) | ResNet-18 | 3.5 | 61.9M | 50 ×25 | 50×25 |
| LaneSegNet + P-MapNet (SDMap cross-attention) | ResNet-18 | 3.3 | 61.4M | 100 ×50 | 100×50 |
| Ours (LaneSegNet + Spatial Enc.) | ResNet-18 | 3.7 | 56.6M | 200×100 | 200×100 |
| Ours (LaneSegNet + Spatial Enc. + Tokenization) | ResNet-18 + Transformer | 3.6 | 59.9M | 200×100 | 200×100 |
| Ours (LaneSegNet + Spatial Enc. + Tokenization + TGD) | ResNet-18 + Transformer | 3.3 | 67.0M | 200×100 | 200×100 |

In Table 5, we report the inference speeds and model parameters. Our model utilizes a lightweight ResNet-18 (13M parameters) to extract SD features for the map spatial encoding component and directly add the SD feature to the BEV feature. The increased latency is primarily attributed to the CNN-based SDMap encoder. P-MapNet uses cross-attention to fuse the 2D-grid based SDMap feature with BEV queries, the complexity of which is proportional to $O(h_{bev} * w_{bev} * h_{SD} * w_{SD})$. If their resolutions are large, such as $200 \times 100$ in LaneSegNet, it will consume much more GPU memory and reduce computing efficiency. Thus it is necessary to downsample the BEV feature and SD feature before fusing them via cross-attention. Despite downsampling, the inference speeds and performances of P-MapNet still lag behind our model with spatial encoding and SD add operation. The map tokenization introduces several Transformer self-attention and cross-attention layers (3.2M parameters), and the fusion computational complexity is $O(h_{bev} * w_{bev} * N_{SD})$, where $N_{SD}$ is the maximum number of SDMap elements in a batch and $N_{SD} << h_{SD} * w_{SD}$.

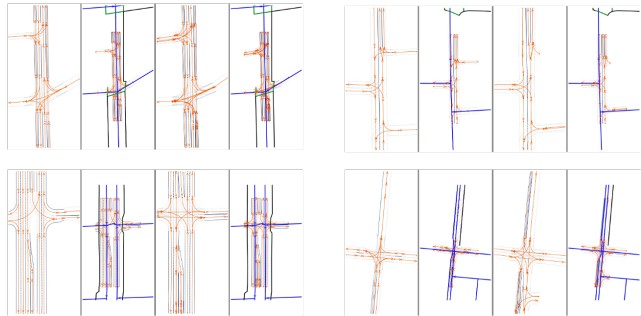

Figure 5: Visualization results on some cases that the given SDMaps has some inconsistency with the lane annotations. For each example, we show 4 sub-figures: GT lane segments, GT lane segments with SDMap, predicted lane segments and predicted lane segments with SDMap.

We also test our model on the Jetson Orin X platform using ONNX deployment, using the SD fusion module. Under FP16, the inference latency for the spatial encoding and SD feature addition is approximately 2 ms. Combining both spatial encoding and tokenization-based cross-attention does not exceed 4 ms. Such performances can meet the requirements for real-time performance for auto-driving vehicles.

### 4.5 QUALITATIVE RESULTS

In Figure 4, we present a comparison between the predicted lane segment results of our proposed model and LaneSegNet. Overall, our model demonstrates superior accuracy in predicting lane geometry and topology. The predicted lane directions align closely with the SDMap road lines. However, LaneSegNet faces challenges in detecting key intersections and long-distance lane lines due to less prominent visual features. In contrast, our model successfully detects junctions and lanes in long-range scenarios, thanks to the complementary SDMap feature.

In certain challenging cases, we notice inconsistencies between the lane annotations and the SDMap road lines, as illustrated in Figure 5. Some lanes are annotated in the map without corresponding SDMap road lines, while other SDMap lines lack corresponding annotated lanes. The presence of such inconsistent annotations necessitates our model to strike a balance between predictions derived from visual features and SDMap features.

## 5 DISCUSSION

In this work we propose to incorporate SDMap information as prior to enhance the predictions of geometry and topology in the lane segment perception. We conduct two complementary methods to encode the geometry and topology information in SDMap and pre-fuse SD feature and tokens into the BEV feature. To further explore the mutual relationships between the geometrical and topological feature, we design a topology-guided decoder to iteratively optimize both geometry and topology. The experiments validate the effectiveness of two combined encoding methods and the proposed topology-guided decoder. We also study the effect of SDMap noise on the performance considering real-world practical applications. Our model achieves state-of-the-art performance on the OpenLaneV2 dataset.

**Limitation.** While SDMaps offer valuable information regarding the geometry and topology of road structures, the information is currently restricted to the road level, lacking lane-level attributes. In addition, the discrepancy between SDMaps and the actual visual environment pose challenges for the perception model in practical applications. Moreover, SDMap may contain errors of several meters due to the positioning shifting and their inherent ambiguity. Future works should focus more on improving the quality of SDMaps and increasing the robustness of the model when the maps are inconsistent with real environments.

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

## A  SDMap Encoding

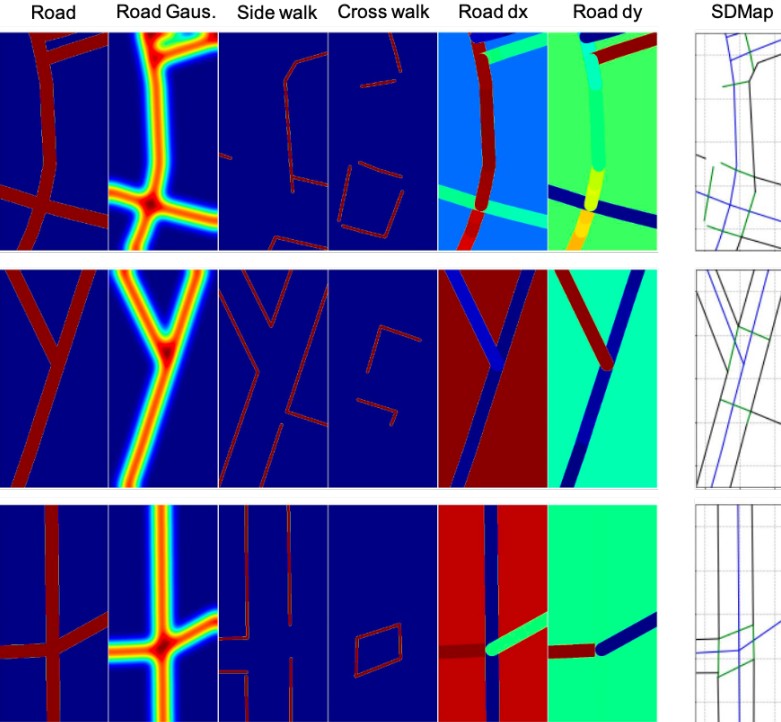

Figure 6: Visualization results on the map spatial encoding method. The 1-th map: the road map; the 2-th map: the road map with Guassian Blurred; the 3-th map: the side walk map; the 4-th map: the cross walk; the 5-th map: the dx map (cosine of the inclination angle) of the road line segments; the 6-th map: the dy map (sine of the inclination angle) of the road line segments.

For the spatial map encoding method, it has 6 channels of 2D canvas maps to represent different road types and attributes of elements in SDMap. As shown in Figure 6, the shape map of roads, the Gaussian blurred shape of roads, the shape map of cross walks, the shape of side walks, and the cosines and sines of the inclination angle of the road line segments that express the curvature of the roads. The map size is set to $800 \times 400$, which is 4 times to the size of BEV feature grids $200 \times 100$. So each grid in the encoded spatial map corresponding $0.125m \times 0.125m$ in actual BEV range. We assume the width of the road in each map are set to 6m and the widths of the cross walk and side walk are set to 1.25m by default. Then we use a lightweight ResNet-18 He et al. (2016) without pre-training to extract 2D feature from these canvas maps. The strides of the four stages of ResNet-18 are set to [2, 2, 1, 1]. As a result, the output feature are $4\times$ downsampled w.r.t the original input, having the same size with the BEV feature, i.e., $200 \times 100$.

## B  Model architecture

**Implementation details.** For the BEV feature extractor, we follow LaneSegNet Li et al. (2024) to adopt the BEVFormer-like architeure. It use ResNet50 He et al. (2016) and FPN Lin et al. (2017) for multi-scale image feature extraction and aggregation. The number of BEV Encoder layers is set to 3. The size of BEV feature grids is set to $200 \times 100$, corresponding to $\pm 50m$ and $\pm 25m$ in the x and y directions. For the decoder part, the number of query is set to 200. The number of decoder layers is set to 6.

We conduct all training experiments on 8 Tesla A100 GPUs. When training, we employ the AdamW Loshchilov & Hutter (2018) as the optimizer. The initial learning rate is set to $2e^{-4}$ with a cosine annealing schedule. All experiments are conducted with a total batch size of 8 for 8 GPUs and a total training epochs of 24.

**Training loss.** Regarding the loss, we combine mainly four types of loss, regression loss, calssification loss, segmentation loss and topology loss:

$$\mathcal{L} = \lambda_{reg}\mathcal{L}_{reg} + \lambda_{cls}\mathcal{L}_{cls} + \lambda_{seg}\mathcal{L}_{seg} + \lambda_{top}\mathcal{L}_{top} + \lambda_{type}\mathcal{L}_{type}, \tag{2}$$

where $\mathcal{L}_{reg}$ means L1 Loss for regressing location of each instance, $\mathcal{L}_{cls}$ supervises each instance category of left boundary, right boundary and centerline by Focal Loss, $\mathcal{L}_{seg}$ contains traditional Cross Entropy Loss and Dice Loss for segmentation tasks and $\mathcal{L}_{top}$ uses Focal Loss for topology connection. $\mathcal{L}_{type}$ applies cross-entropy loss on the classification of laneline types between $\{\hat{a}_l, \hat{a}_r\}$ and $\{a_l, a_r\}$ correspondingly. The hyperparameters are defined as: $\lambda_{reg} = 0.05$, $\lambda_{cls} = 1.5$, $\lambda_{seg} = 3.0$, $\lambda_{top} = 5.0$, $\lambda_{type} = 0.01$.

**SDMap testing noise levels**. In Figure 3, we compare the model performances under different SDMap noise levels, from level 0 to level 8. The configurations from level-0 to level-8 are: no_noise, rot5_std2_prob0.5, rot5_std5_prob0.5, rot5_std7_prob0.5, rot5_std10_prob0.5, rot5_std20_prob0.5, rot5_std30_prob0.5, rot5_std20_prob1, rot5_std30_prob1.

## C    METRICS

Following LaneSegNet Li et al. (2024), we use the defined lane segment distance to measure the average precision of the detected lane segments. The lane segment distance is defined as a weighted sum of distances between left/right lane boundaries and centerlines and their direction:

$$\mathcal{D}_{ls}(\hat{\boldsymbol{v}}, \boldsymbol{v}) = 0.5 \cdot [\text{Chamfer}\left([\hat{\boldsymbol{v}_l}, \hat{\boldsymbol{v}_r}], [\boldsymbol{v}_l, \boldsymbol{v}_r]\right) + \text{Frechet}\left(\hat{\boldsymbol{v}_c}, \boldsymbol{v}_r\right)]. \tag{3}$$

Based on this distance metric, the average precision, $\text{AP}_{ls}$, is computed over three matching thresholds: 1.0m, 2.0m, 3.0m. The $\text{AP}_{ped}$ is based on the Chamfer distance to evaluate the non-directional pedestrian crossing, with thresholds of 0.5m, 1.0m, and 1.5m for evaluation.

Similar to $\text{TOP}_{ll}$, $\text{TOP}_{lsls}$ represents the similarity between the predicted lane graph among lane segments and the ground truth. It is defined as the averaged vertice mAP between the ground truth $\mathcal{G} = (V, E)$ and the predicted graph $(\hat{V}', \hat{E}')$:

$$\text{TOP} = \frac{1}{|V|} \sum_{v \in V} \frac{\sum_{\hat{n}' \in \hat{N}'(v)} P\left(\hat{n}'\right) \mathbf{1}_{\text{condition}}\left(\hat{n}' \in N\left(v\right)\right)}{|N\left(v\right)|}, \tag{4}$$

where $N(v)$ denotes the ordered list of neighbors of vertex $v$ in the ground truth ranked by confidence and $P(v)$ is the precision of the $i$-th vertex $v$ in the predicted ordered list. The $\text{TOP}_{lsls}$ is for topology among lane segments on the graph $(V_{ls}, E_{lsls})$, while the $\text{TOP}_{lste}$ is for topology between lane segments and traffic elements on the graph $(V_{ls} \cup V_{te}, E_{lste})$.

Besides, we also report the results on the performances on the multiple tasks of OpenLaneV2 map element bucket in Table 2, with extra metrics of $\text{DET}_t$, and $\text{TOP}_{lste}$. The $\text{DET}_t$ is to evaluate the task of traffic element detection, which is based on IoU distance between the detected traffic element boxes and the ground truth boxes and is averaged over different traffic element attributes. The $\text{TOP}_{lste}$ is to evalaute the task of topology prediction between lane segments and traffic elements.

## D    MORE ABLATION RESULTS

**Ablation on the topology head**. We present the results of the ablations on the design of the topology head. LaneSegNet Li et al. (2024) firstly uses two MLPs to project the instance queries $\mathcal{Q} \in \mathbb{R}^{N \times D}$ to two embeddings $\boldsymbol{E}_1 \in \mathbb{R}^{N \times D_e}$ and $\boldsymbol{E}_2 \in \mathbb{R}^{N \times D_e}$, and then broadcast both embeddings to new shapes of $\mathbb{R}^{N \times N \times c}$. Finally, two embeddings with shape of $\mathbb{R}^{N \times N \times D_e}$ are concatenated at the feature dimension to form a shape of $R^{N \times N \times 2*D_e}$ and sent to an association MLP to predict the adjacent matrix with shape of $R^{N \times N \times 1}$. In the implementation of the proposed connect head, we also use two MLPs to project the queries to two embeddings $\boldsymbol{E}_s \in \mathbb{R}^{N \times D_e}$ and $\boldsymbol{E}_e \in \mathbb{R}^{N \times D_e}$, but we simply compute their inner-products as the adjacent matrix among different lane segment instances. As shown in Table 6, with fewer parameters, the topology head via inner-product computing has achieved similar result w.r.t the mAP metric and better result w.r.t the topology metrics in comparison to the association MLP.

Table 6: Ablation on the topology head.

| Topology Head | #Params. | mAP | $AP_{ls}$ | $AP_{ped}$ | $TOP_{lsls}$ |
|---|---|---|---|---|---|
| Association MLP | 321k | 37.0 | 34.5 | 39.5 | 28.5 |
| Inner Product | 129k | 36.8 | 34.6 | 39.1 | 28.9 |

**Ablation on the fusion position for SDMap**. As SDMaps provide road-level rather than lane-level geometry and topology, there inevitably existing meter-level errors or inconsistent road description. Thus it is critical to choose an appropriate position to fuse the SD information into the neural network model. From the view of fusion position, we classify the SDMap fusion into two categories: SD fusion in the BEV encoder and SD fusion in the lane segment Decoder. In the Section 4.2, we make ablations on fusing SD features on the BEV queries or BEV features. In this part, we expore to fuse the SDMap in the lane segment Decoder part.

In each layer of the lane segment Decoder, we insert an additional SD cross-attention layer between the self-attention layer and the lane-attention cross-attention layer. Note that we do not use the topology-guided self-attention for the sake of reducing the effects from other variables. The lane segment instances queries are interacted with the SD feature $F_s \in \mathbb{R}^{d \times h \times w}$ (as keys and values) through this SD cross-attention layer. As shown in Table 7, fusing SD features in the lane segment decoder, performs worse than fusing SD features in the BEV Encoder part regardless of in BEV quries or BEV features. This phenomenon suggests that the geometric and topological information represented in SDMap is inherently coarse, rendering it unsuitable for fusion near the output of the model. Instead, it is better suited for fusing in the earlier stage of the model as a coarse prompt of road structure.

Table 7: Ablation on the fusion position of SD features.

| SDMap feature Fusion Position | mAP | $AP_{ls}$ | $AP_{ped}$ | $TOP_{lsls}$ |
|---|---|---|---|---|
| BEV Encoder (BEV Query Fusion) | 38.3 | 37.2 | 39.4 | 31.8 |
| BEV Encoder (BEV Feature Fusion) | 39.1 | 37.3 | 40.9 | 30.7 |
| Lane segment Decoder (Instance Query Fusion) | 37.9 | 36.8 | 39.1 | 30.5 |

# E    MORE VISUALIZATION RESULTS

The Figure 7 show more visualization examples and the comparisons with LaneSegNet. See more examples in the supplementary materials. All the visualization results of LaneSegNet is based on the official release weight [1].

---

[1] https://huggingface.co/OpenDriveLab/lanesegnet_r50_8x1_24e_olv2_subset_A/resolve/main/lanesegnet_r50_8x1_24e_olv2_subset_A.pth

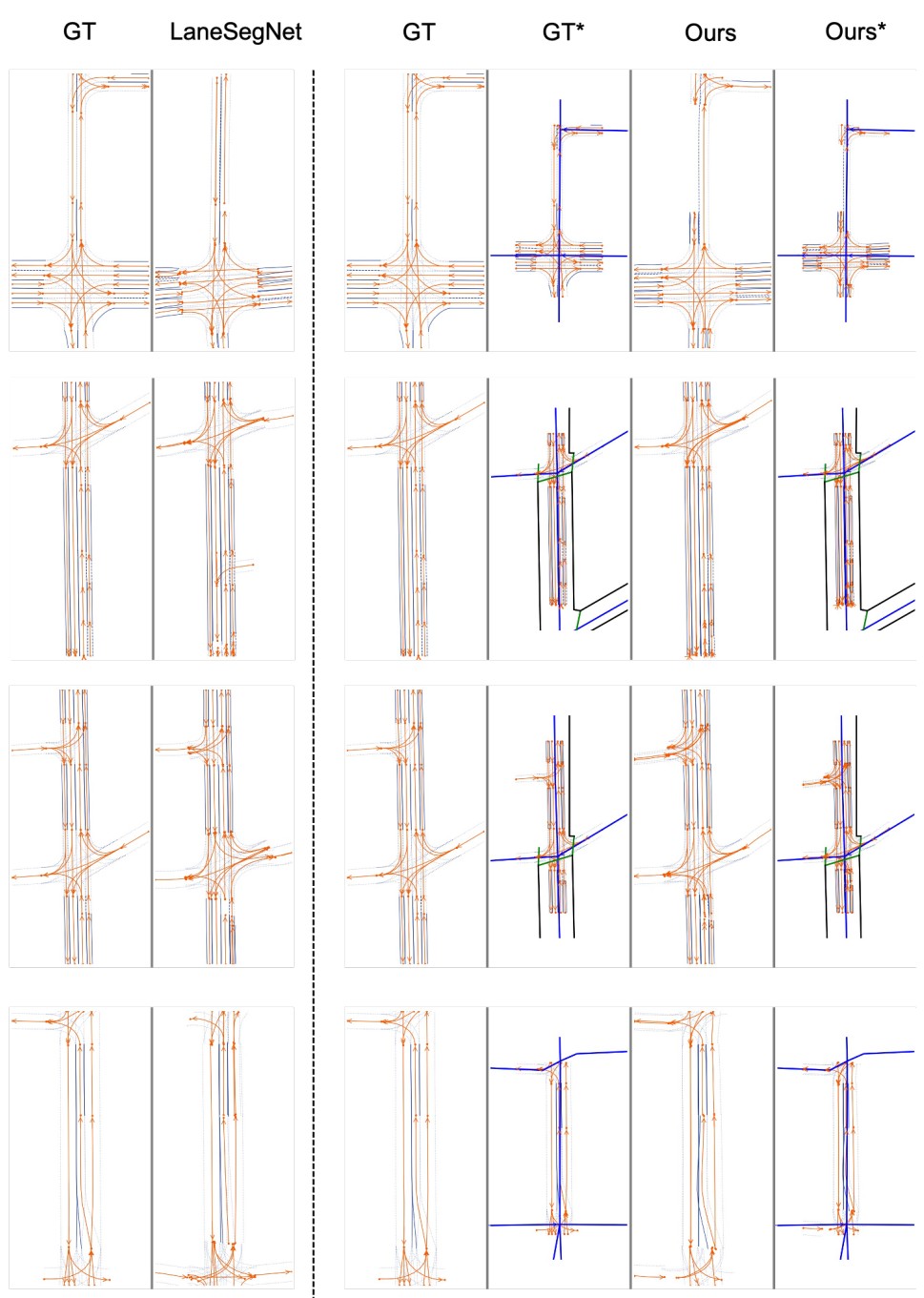

Figure 7: Visualization results. * means the lane segments plotted with the SDMap elements.

