# OpenReview forum: "TopoSD: Topology-Enhanced Lane Segment Perception with SDMap prior"
_ICLR.cc/2025/Conference — Submitted to ICLR 2025_

### Official Review · Reviewer_Drcu · 2024-10-29

**Soundness:** 2
**Presentation:** 1
**Contribution:** 2
**Rating:** 5
**Confidence:** 4

**Summary:**

As annotating HD maps is expensive for real-world applications, researchers have started generating mapping elements based on onboard sensors on self-driving vehicles. The authors propose a method to integrate SD maps as prior knowledge for better generation performance. The contribution of this paper is threefold: (1) two complementary SD maps encoding methods are introduced; (2) a Topology-Guided Decoder is proposed to better leverage geometrical and topological features; (3) achieving SOTA performance on the OpenLane-V2 dataset. The two  SD map encoding methods refer to [spatial map encoding] processing SD map elements drawn on various canvases via CNN and [map tokenization] encoding SD map elements via one-hot encoding and Transformer. The Topology-Guided Decoder (TGD) refers to modifying deformable attention modules to let predicted topology information influence the prediction of geometric information. Experimentally, the proposal method achieves better performance than the baseline method, LaneSegNet, and some other methods, like SMERF and P-MapNet.

**Strengths:**

- The authors conduct detailed ablation studies to demonstrate the effectiveness of the proposed modules.
- Concerning real-world applications, the authors conduct analysis on the effect of noisy SD map data on model performance.

**Weaknesses:**

- In line 15, the authors state that '... long-range perception due to the limited field of view of cameras.' This should refer to the distance a camera can 'see'. The norn of 'field of view' should refer to the angular extent of the camera. The authors should rephrase this sentence for clarity.
- Grammar errors are common, such as 'surround view' -> 'surrounding view' in line 175, 'local aligned' -> 'locally aligned' in line 175,  'forms, i.e.' -> ' forms, i.e.' in line 177, etc. The authors should use tools like Grammarly to complete a grammar check.
- The first two contributions of the paper are limited. The proposed modules to encode SD map information are straightforward and commonly seen, such as in UniHDMap [1] and MapVison [2]. The TGD module depends on the learning results of the deformable attention mechanism.


[1] Kou, Genghua, et al. "UniHDMap: Unified Lane Elements Detection for Topology HD Map Construction."

[2] Yang, Zhongyu, et al. "MapVision: CVPR 2024 Autonomous Grand Challenge Mapless Driving Tech Report." arXiv preprint arXiv:2406.10125 (2024).

**Questions:**

- Please carefully check grammar.
- To verify the novelty of the SD map encoding modules, it is suggested that the proposed method should be compared to other encoding methods, both theoretically and experimentally.
- To verify the novelty of the TDG module, it is suggested that ablation studies on the design choice be done and that the proposed method be logically compared to other design choices.

---

> ### Author Response · Authors · 2024-11-19
> **Author response to Reviewer Drcu**
>
> We appreciate your valuable comments and questions. We hope that our response can address your concerns.
>
> > ***Q1: In line 15, the authors state that '... long-range perception due to the limited field of view of cameras.' This should refer to the distance a camera can 'see'. ... The authors should rephrase this sentence for clarity***
>
> A1: Thank you for your suggestion. Here, what we aim to convey is that the views of onboard cameras are limited because of their installation angles and positions, which restrict their perception range to the vehicle’s immediate surroundings and hinder their ability to capture long-range structural information about the road. In contrast, SDMaps provide road structure information in a bird’s-eye view, enabling a broader view of the environment. We will rephrase this sentence for clarity.
>
> > ***Q2: Grammar errors are common, such as 'surround view' -> 'surrounding view' in line 175 ...***
>
> A2: Sorry for such grammar errors. We would carefully check and revise all typos and grammar errors in the paper.
>
> > ***Q3:  The first two contributions of the paper are limited ... & the novelty of the SD map encoding modules & the novelty of the TDG module***
>
> A3: Here we would like to clarify our contributions. Broadly, SDMap encoding methods can be categorized into spatial encodings, tokenization encodings, and other variations, while fusion strategies typically involve approaches like cross-attention, addition, or concatenation. In our work, we combine the advantages of these existing methods and propose a novel spatial map encoding that integrates multiple attributes, including shape, type, and curvature. This approach achieves a balance between computational efficiency and complementary performance enhancements.
>
> Regarding the works UniHDMap [1] and MapVision [2], as you noted, they also employ SMERF-like SDMap encoding and fusion methods. To demonstrate the effectiveness of our method in the paper, we conducted comprehensive comparisons with LaneSegNet + SMERF and LaneSegNet + P-MapNet in terms of both performance (Table 1) and inference speed (Table 5), showing the advantages of our approach.
>
> The core idea of the proposed TGD module is to insert the topology head into each decoder layer. This allows the module to utilize the successor and predecessor relationships predicted in the adjacency matrix, enabling it to iteratively refine instance-level features from the previous layer. In Appendix D, we conduct ablations on the choices of the topology prediction head.
>
> [1] Kou, Genghua, et al. "UniHDMap: Unified Lane Elements Detection for Topology HD Map Construction."
>
> [2] Yang, Zhongyu, et al. "MapVision: CVPR 2024 Autonomous Grand Challenge Mapless Driving Tech Report." arXiv preprint arXiv:2406.10125 (2024).

---

> > ### Author Response · Authors · 2024-11-26
> > **A Gentle Reminder of the Final Feedback**
> >
> > Dear reviewer,
> >
> > We would be grateful if we could get some feedback from you about the raised concerns. If there are any remaining questions or points requiring clarification, we are happy to address them before the discussion deadline. Please also consider updating the score if all concerns are addressed.
> >
> > Best，
> >
> > The authors of Paper #7200

---

> > > ### Comment · Reviewer_Drcu · 2024-11-29
> > >
> > > Thanks you for your response. No further comment.

---

### Official Review · Reviewer_ckWD · 2024-10-31

**Soundness:** 2
**Presentation:** 3
**Contribution:** 3
**Rating:** 5
**Confidence:** 4

**Summary:**

This paper proposes a novel method called TopoSD to enhance the online generation of high-definition maps (HDMaps) using prior knowledge from standard definition maps (SDMaps). The model processes perspective images captured by cameras arranged in a surround-view configuration, augmenting the online prediction capabilities of the long-range HDMap with locally aligned SDMaps.

**Strengths:**

1. The paper is well-structured, demonstrating strong logical coherence, and the definitions of geometric and topological tasks are presented clearly and comprehensively.
2. Quantitative experiments conducted on the OpenlaneV2 dataset demonstrate significant performance improvements and provide valuable insights for result interpretation.
3. The framework exhibits superior real-time performance while constructing a map at a distance, outperforming related works.

**Weaknesses:**

1. The definition of SDMap in the article and its acquisition method during the experiment are overly concise and ambiguous. Further clarification regarding the application of SDMap within the framework is necessary.
2. Section 4.3 lacks comprehensive error analysis in practical scenarios. For instance, when generating an SDMap, in addition to positional offsets, the article should consider other potential inaccuracies.
3. The article does not provide a detailed analysis of the performance enhancements attributable to SDMap in the modeling process.

**Questions:**

1. Could you offer a clearer and more comprehensive definition of SDMap along with its specific application within the framework? Why not use some open-source maps like OpenStreetMap?
2. Does the framework require addressing the alignment of structured information in contiguous areas of SDMap before it is input into the network? If so, what methods are employed in the framework to achieve this?

---

> ### Author Response · Authors · 2024-11-19
> **Author response to Reviewer ckWD**
>
> We appreciate your valuable comments and questions. We thank you for the positive comments. We hope that our response can address your concerns.
>
> > ***Q1: The definition of SDMap within the framework & Why not use some open-source maps like OpenStreetMap?***
>
> A1: We thank you for your valuable suggestions. Within our framework, the concept of SDMap is synonymous with the navigation map. While different map providers may have slight variations in their SDMap formats, they share many similarities, and our solution is designed to leverage these commonalities effectively.
>
> The core of SDMap or navigation map is to provide the basic **road-level** geometry and topology information for navigation, such as the centerlines of the roads. The term "standard-definition map" is relative to high-definition (HD) maps (https://en.wikipedia.org/wiki/High-definition_map). HD maps offer precise **lane-level** geometric and topological details, typically with centimeter-level accuracy. In contrast, SDMaps may have meter-level inaccuracies. In our work, the SDMaps we use come from the annotations of OpenLanev2. It also defines SDMap (https://github.com/OpenDriveLab/OpenLane-V2/blob/master/docs/features.md#sd-map).
>
> In previous works such as P-MapNet and SMERF, SDMap does not have a specific or uniform definition. However, in summary, SDMaps outline road-level geometry and topology. This contrasts with HDMaps, which offer comprehensive semantic and geometric lane-level details.
>
> We did not use OpenStreetMap mainly because we selected the OpenLaneV2 lane segmentation perception task as our benchmark. It provides a well-defined and accurate SDMap annotation, which is more convenient to use. Importantly, this benchmark introduces a new annotation format for lane segments in map learning, going beyond traditional map element detection or centerline perception. It also establishes metrics to evaluate overall performance on lane lines, centerlines, types, and topologies.
>
> Even though the other datasets (e.g. NuScenes) can access SDMap annotations through OpenStreetMap (OSM), there is currently no widely accepted standardized method to align OSM data with the nuScenes and Argoverse2 datasets.
>
> For these reasons, we chose OpenLane-V2 with SDMap annotations as our benchmark and conducted extensive experiments to validate the effectiveness of the SDMap fusion component, aiming for relative improvements over LaneSegNet.
>
> > ***Q2: Section 4.3 lacks comprehensive error analysis in practical scenarios***
>
> A2: Thanks for your comments. In real-world scenarios, SDMap accuracy is mainly affected by both vehicle localization errors and intrinsic inaccuracies in producing SDMaps. These combined errors result in global transformations (translations and rotations) as well as local perturbations of road elements. As directly simulating these individual error sources is challenging, we adopt a simplified approach by adding random global transformations and rotation noise levels to approximate their cumulative effects while amplifying the magnitude of random noise levels (as shown in Figure 3).
>
> > ***Q3: The article does not provide a detailed analysis of the performance enhancements attributable to SDMap in the modeling process***
>
> A3： Thank you for your comment. We are not entirely certain if we have fully understood your concern. If the intent is to request an analysis of the performance gains attributable to each component of the SDMap fusion modules, we have provided such results in Table 3. These results validate the effectiveness of each component, demonstrating that our SDMap fusion modeling approach achieves significant performance improvements over the baseline model.
>
> > ***Q4: Does the framework require addressing the alignment of structured information in contiguous areas of SDMap before it is input into the network? If so, what methods are employed in the framework to achieve this?***
>
> A4：We would like to highlight that SDMaps, as mentioned above, may not be perfectly aligned with real-world road environments when transformed into the ego frame due to inherent vehicle localization errors and intrinsic inaccuracies in their generation. As illustrated in Figure 6, when SDMap elements are encoded into spatial map representations using rasterization, the polylines of the SDMaps are drawn on the canvas with a certain width, and Gaussian blur is applied to represent their ambiguity. This process inherently introduces quantization errors, which can lead to minor yet reasonable inaccuracies when integrating them into the grid-based BEV feature. In other words, SDMaps provide coarse road-level information and we coarsely fuse such information as well. For the map tokenization, the precise coordinates are encoded into SD tokens. In such a way, we expect to use the attention-based mechanism to adaptively filter those SDMap elements that do not align well with the visual feature.

---

> > ### Author Response · Authors · 2024-11-26
> > **A Gentle Reminder of the Final Feedback**
> >
> > Dear reviewer,
> >
> > We thank you for your thoughtful review and hope our responses have addressed your concerns.  If there are any remaining questions or points requiring clarification, we are happy to address them before the discussion deadline.  Please also consider updating the score if all concerns are addressed.
> >
> > Best，
> >
> > The authors of Paper #7200

---

> > ### Comment · Reviewer_ckWD · 2024-11-26
> >
> > All concerns have been addressed, no further comments. Thank you for your response.

---

### Official Review · Reviewer_sYjC · 2024-11-01

**Soundness:** 3
**Presentation:** 2
**Contribution:** 2
**Rating:** 5
**Confidence:** 5

**Summary:**

This work proposed an online mapping method named TopoSD, which enhances lane segment perception capabilities with SDMap priors. Concretely, SDMap elements are encoded into spatial map representations with CNN and instance tokens with transformer encoder and fused with BEV features at different stages. The design mainly lies in the fusion of SDMap priors, while the main structure still follows LaneSegNet. Multiple heads are concatenated to enable the model to simultaneously predict various elements that are required by the online road map. Experiments are conducted on the OpenLane-v2 dataset and demonstrate a significant performance gain compared to baselines. Besides, TopoSD also shows robustness to SDMap noises, which enhances its real-world application values.

**Strengths:**

- SDMap is a much more easily accessible map prior compared to HDMap and shows the basic structures of a road network. The introduction of it is intuitive and of great practical value.
- The fusion of BEV feature and SDMap priors at different levels is simple but effective, leading to a significant performance gain, as demonstrated in the experiments.
- The study on the effect of mis-aligned SDMap is novel, which is a common case due to the SDMap collection methods.

**Weaknesses:**

- Although the metrics have been elevated greatly in OpenLane-V2, the generated online map still seems very terrible and contains **lots of** significant errors, overlaps, and wrong detections, as displayed in the qualitative results on Page 9. It prevents TopoSD from being put into real use.
- Although the study on the influence of SDMap error is novel, the experimental results seem contradictory to the claims TopoSD proposes, which makes this section of study ill-defined. Since the TopoSD is robust to the influence of SDMap deviation or rotation, how much SDMap contribute to the perception result of TopoSD? Besides, there seems no specific design to rectify the SDMap prior errors in your model design.
- As shown in Table 4, the performance of TopoSD trained with SDMap noise is even worse than the baseline LaneSegNet without any SDMap priors. Such a kind of robustness is far from satisfactory.

**Questions:**

- Why is SDMap’s range $\pm100m \times \pm50m$ while the perception range still remains $\pm50m\times\pm25m$? As far as I am concerned, to truly reflect model’s long range perception performance, the perception range should also be extended to the same range as SDMap’s. Could you also provide experiment results under this setting?
- Could you also provide the results of TopoSD with ResNet-50 backbone, which is the original setting of LaneSegNet and more commonly compared with in the context of online mapping?

---

> ### Author Response · Authors · 2024-11-19
> **Author response to Reviewer sYjC**
>
> We appreciate your valuable comments and questions. We thank you for the positive feedback and the focus on real applications. We hope that our response can address your concerns.
>
> > ***Q1： Although the metrics have been elevated greatly in OpenLane-V2, the generated online map still seems very terrible and contains lots of significant errors, overlaps, and wrong detections, as displayed in the qualitative results on Page 9***
>
> A1:   Thank you for your careful observations. We acknowledge that the visualized results on Page 9 are far from perfect with some overlaps and incorrect detections. However, compared to the baseline LaneSegNet, the utilization of SDMap demonstrates significant overall improvements in lane detection accuracy as well as long-range perception and recognition of key geometry and road topology. It's important to note that we trained our model on 27,000 samples and tested it on 4,800 samples, following the LaneSegNet approach for a fair comparison. The number of training data samples is small and has not yet reached the scale necessary to fully represent real-world autonomous driving scenarios. We believe that by scaling the training data to a substantial level, the model will be able to mitigate most cases of incorrect detection errors.
>
> > ***Q2: Although the study on the influence of SDMap error is novel, the experimental results seem contradictory to the claims TopoSD proposes***
>
> A2: We appreciate your insightful comments. Our study on the influence of SDMap error aims to reveal the potential problems of using nearly perfectly accurate SDMaps as input for evaluating models with SDMap fusion. Thus, we investigate how these models perform under conditions involving noisy SDMaps. However, the testing conditions are extreme to test their robustness which may not correspond to the errors in real applications. In such challenging hand-designed conditions, we expect models to adaptively rely more on visual features while minimizing interference from inaccurate SDMap data as much as possible. These experiments reveal potential vulnerabilities in models trained without SDMap noise augmentation.
>
> In practical applications, we believe combining large-scale training data with SDMap data augmentation can probably bring stable improvements over the model trained without using SDMap input. Of course, to deploy such models to real applications, practitioners must also strive to ensure the input navigation maps (or SDMap) do not have significant errors as input data consistency inherently benefits performance.
>
> Regarding model architecture, we haven't implemented specific designs to address this issue. We think large-scale training data with data augmentation can manage it. And we hope future research can explore architectural solutions to this challenge.
>
>
> > ***Q3：As shown in Table 4, the performance of TopoSD trained with SDMap noise is even worse than the baseline LaneSegNet without any SDMap priors***
>
> A4：We apologize for the misunderstanding caused by the writing error. To clarify, in Table 4, LaneSegNet's mAP is 33.5 rather than 35.5. Our TopoSD model, even when trained with noisy SDMap input, still outperforms the baseline LaneSegNet in terms of both mAP and TOP$_{lsls}$ metrics under noisy test conditions.
>
> More significantly, the key focus of Table 4 is to examine the relative performance degradation between models trained with and without our proposed data augmentation strategy. While performance drops are inevitable when dealing with inherently inconsistent input modalities, our model, trained with the noisy SDMap augmentation technique, demonstrates superior robustness by effectively minimizing this performance degradation.
>
>
>
> > ***Q4: Why is SDMap’s range $±100m \times ±50m$  while the perception range still remains  $±50m \times ±25m$ ?***
>
> A4: It's worth noting that the larger range of SDMap range only works for the map tokenization as we encode its coordinates to the SD tokens. For the spatial encodings, due to that we adopt the same size of the BEV feature to encode the SD feature, the real range of the SD feature is consistent with BEV size. In particular, we do not change the original BEV perception and the BEV feature resolution because the lane annotations are still restricted within $±50m \times ±25m$. Though we can conduct experiments with larger BEV ranges, the restricted coverage of lane line annotations makes it difficult to quantify the advantages.
>
> > ***Q5: Could you also provide the results of TopoSD with ResNet-50 backbone***
>
> A5: For a fair comparison with LaneSegNet, we indeed use the same ResNet-50 backbone for image feature extraction. For processing the encoded spatial SD maps, we employ a lighter ResNet-18 architecture as our spatial SDMap encoder.

---

> ### Author Response · Authors · 2024-11-26
> **A Gentle Reminder of the Final Feedback**
>
> Dear reviewer,
>
> We would sincerely appreciate it if we could get some feedback from you regarding the above concerns. If there are any remaining questions or points requiring clarification, we are happy to address them before the discussion deadline.  Please consider raising the score if all concerns are addressed.
>
> Best，
>
> The authors of Paper #7200

---

> > ### Comment · Reviewer_sYjC · 2024-11-27
> >
> > Thanks for the author's response to my questions. However, I still remain doubtful about the experiments about the influence of SDMap error. Since you don't have a corresponding design to mitigate this issue, I am not convinced of the necessity and rationale behind this part. Maybe a better writing logic here can help. I will accordingly raise my score.

---

> ### Author Response · Authors · 2024-11-27
> **Author response to Reviewer sYjC**
>
> Thank you for your feedback.
>
> In addressing the potential negative effects of SDMap errors, we acknowledge that this paper does not resolve the issue from the perspective of model design. Instead, our focus is to highlight that, under the current evaluation framework, training a model with highly accurate SDMap input may not adequately reflect the model's robustness to SDMap errors. To address this, we evaluate models using noisy SDMap inputs and propose a straightforward yet effective training strategy based on SDMap data augmentation. As demonstrated in Table 4 and Figure 3, this approach effectively mitigates the issue.
>
> On another level, it is also essential to consider how SDMap, as auxiliary information, can truly provide benefits. Our strategy is to reduce the model's dependence on the precise geometry of the SDMap and instead utilize it to enhance the understanding of overall road structure and approximate geometry. The primary source of information for map prediction should remain the visual features.
>
> In real-world applications, we believe that future works can focus on addressing these challenges through improvements in model design or data quality. We thank you for your thoughtful question and hope our response provides clarity.

---

> ### Author Response · Authors · 2024-11-28
> **Author response to Reviewer sYjC**
>
> Here, we would like to further elaborate on the rationale behind the proposed SDMap noise augmentation strategy:
>
> By introducing random noise to the polylines of SDMap elements in each sample pair, the model is exposed to diverse training samples rather than memorizing fixed relationships between the surrounding images and SDMaps. This enhances the robustness against the (global) SDmap error by encouraging the model to extract relative relations among elements in SDMap. Thus the model can learn key road structures, such as curvature, and topological connections from SDMap, rather than just "copying and pasting" the geometry of SDMap to generate final results.
>
> In the community of deep learning, injecting Gaussian noise into inputs is also a common regularization technique to enhance neural network robustness. For instance:
> - In image classification tasks ([1], [2], [3]), adding Gaussian or random noise to input images has been shown to improve model accuracy and robustness.
> - In adversarial training ([4]), randomly adding noise to each pixel strengthens the model's resilience against adversarial examples.
>
> Our results also demonstrate that this noise augmentation strategy effectively enhances the model's robustness, as shown in Tab 4 and Fig 3. Thanks for your review and we are willing to address any remaining concerns.
>
> References:
>
> [1] Lopes, Raphael Gontijo, et al. "Improving robustness without sacrificing accuracy with patch gaussian augmentation." arXiv preprint arXiv:1906.02611 (2019).
>
> [2] Zhong, Zhun, et al. "Random erasing data augmentation." Proceedings of the AAAI conference on artificial intelligence. Vol. 34, No. 07. 2020.
>
> [3] Cubuk, Ekin D., et al. "Randaugment: Practical automated data augmentation with a reduced search space." Proceedings of the IEEE/CVF conference on computer vision and pattern recognition workshops. 2020.
>
> [4] Goodfellow, Ian J., Jonathon Shlens, and Christian Szegedy. "Explaining and harnessing adversarial examples." arXiv preprint arXiv:1412.6572 (2014).

---

### Official Review · Reviewer_D5MY · 2024-11-03

**Soundness:** 3
**Presentation:** 3
**Contribution:** 3
**Rating:** 5
**Confidence:** 4

**Summary:**

This paper focuses on the task of online map generation. In order to improve the performances of online map construction, the authors adopt SDMaps as prior to enhance BEV feature. To incorporate the SDMaps prior with BEV-based framework, the authors introduce two distinct encoding methods: (1) spatial map encoding and (2) map tokenization. The spatial map encoding is added into the initial BEV query and the SDMaps tokens are used as key and values in the cross attention of BEV encoder. Additionally, to improve the performances of topology prediction, the authors proposes a topology-guided self attention mechanism to aggregate features of predecessor and the successor. The proposed method achieve state-of-the-art performance on the OpenLaneV2 benchmark.

**Strengths:**

1. The writing and presentation of this paper is good.
2. The authors provide detailed ablation studies to show how the proposed SDMap prior fusion and topology-guided decoder improve the performances.
3. The authors recognize the noise issue in SDMap and mitigate the performance degradation through data augmentation during training.
4. The proposed method achieves high performance compared to recent state-of-the-art methods.

**Weaknesses:**

1. The SDMap Prior Fusion section lacks technical innovation. The authors combine two SDMap representation methods to achieve better results, but both methods are derived from previous works: spatial map encoding from P-MapNet and map tokenization from SMERF. The author should explain the differences between the proposed fusion method and the simply combination of P-MapNet and SMERF (for example: (1) using both spatial map encoding and map tokenization as key\values in cross attention; (2) concat or add spatial map encoding with BEV features and using map tokenization as key\values in cross attention.
2. Some minor writing errors:
(1) In Table 1, Ours-2 achieves lower AP_ped compared to Ours-1. However, the improvement of Ours-2 is 7.2 while Ours-1 is 7.0.
(2) A period is missing before "Similarly" in Line 259.

**Questions:**

1. The authors should explain the technical contributions of their proposed SDMap Prior Fusion and provide a detailed discussion and comparison with P-MapNet and SMERF in the rebuttal. As shown in Table 3, the most significant improvement of SDMap Prior Fusion actually comes from jointly applying Spatial Encoding and Tokenization. I will consider improve my rating if the authors can address my concern.
2. Is separating predecessor and successor in the Topology-guided Self Attention Mechanism the key factor for performance improvement? The author can prodive ablation study through comparing the proposed method with simply aggregating features by adj. matrix without separate predecessor and successor information.

---

> ### Author Response · Authors · 2024-11-19
> **Author response to Reviewer D5MY**
>
> We appreciate your valuable comments and questions. We thank you for the positive comments on this work. We hope that our response can address your concerns.
>
> > ***Q1: The SDMap Prior Fusion section lacks technical innovation***
>
> A1: We thank you for your insightful suggestions. SMERF and P-MapNet are pioneering works that utilize SDMaps to help the BEV perception. From the perspective of SDMap encoding, our work is the first to combine the local and global map representation schemes to achieve complementary advantages. Specifically, the spatial map encoding can describe the local geometry and topology of roads, while the map tokenization of SDMap elements with a Transformer encoder can capture the global relationships. In this work, we study how to combine and where to fuse these two types of SD encodings for better BEV perception. Experimental results indicate that both representations bring complementary improvements without conflict, demonstrating their synergistic effects.
>
> Regarding your first concern about using spatial map encoding and map tokenization as key/values in cross-attention, we conducted separate validation experiments on LaneSegNet, as shown in Table 1. P-MapNet employs cross-attention to fuse SD features into BEV features, with a computational complexity of $O(H_{bev} \times W_{bev} \times H_{SD} \times W_{SD})$. Whereas, the computational complexity of the cross-attention used in our method or SMERF is $O(H_{bev} \times W_{bev}  \times N_{SD})$, where $N_{SD} << H_{SD}\times W_{SD} $. Following LaneSegNet's high-resolution setting (200 x 100), we must downsample both SD and BEV features before cross-attention to reduce computational overhead (this implementation also follows the official code of P-Mapnet). However, this cross-attention approach for spatial encoding fusion not only runs slower than direct feature addition to BEV features and queries (as shown in Table 5) but also yields lower accuracy in the mAP metric compared with the LaneSegNet baseline and our add-based fusion method (Table 1 and 3). Considering that using spatial map encoding as keys/values with downsampling results in a performance drop, we think that employing both types of encodings as keys/values may not be the optimal choice.
>
> Regarding your second concern -- concatenating or adding the spatial map encoding to the BEV feature and using map tokenization as key\values in cross attention, this is indeed the approach we have taken. More specifically, we enhance both BEV queries and BEV features by adding spatial map encoding, a dual-addition strategy that leads to complementary performance improvements, as demonstrated in Table 3.
>
> By the way, in terms of spatial encoding strategy, P-MapNet uses a single channel to represent SDMap polylines, while we employ multiple channels to capture various attributes of the SDMap, such as road shapes, types, and curvature, as shown in Figure 6.
>
> > Q2: ***Some minor writing errors***
>
> A2: We are sorry for these minor writing errors. We would carefully check all typos and writing errors in the paper.
>
> > ***Q3: ... explain the technical contributions of their proposed SDMap Prior Fusion and provide a detailed discussion and comparison with P-MapNet and SMERF in the rebuttal ...***
>
> A3: Thanks for your comments and suggestions. We have explained our technical contributions in the Q1 response. In summary, from the perspective of encoding, there are primarily spatial encodings, tokenization encodings, and others. In terms of fusion, the options mainly include cross-attention, addition, or concatenation. We have combined the advantages of these various methods and introduced a novel spatial position encoding with multiple attributes (shape, types and curvatures). Our approach strikes a balance with moderate computational complexity while providing complementary improvements. Additionally, we investigate the impact of error issues in SDMaps on performance, which is crucial for real-world applications when using SDMaps as supplementary input for autonomous vehicles. We believe that testing the model's stability against noisy SDMaps is essential for approaches that utilize SDMap fusion.
>
> > ***Q4: Is separating predecessor and successor in the Topology-guided Self Attention Mechanism the key factor for performance improvement?***
>
> A4: We are not ensure that we fully understand your statement. Intuitively, the topology and geometric relationships of each lane are primarily influenced by its preceding and succeeding lanes, particularly for their start points and ending points. The adjacency matrix records the preceding and succeeding information between lanes (lane segments or centerlines), where each row represents the succeeding relationships and each column represents the preceding relationships. There may be several choices to use the adj. matrix, but we think aggregating features by the adj. matrix is equivalent to utilizing predecessor and successor information.

---

> > ### Comment · Reviewer_D5MY · 2024-11-23
> >
> > Thanks for your response. All my concerns have been addressed.

---

### Official Review · Reviewer_XgUo · 2024-11-04

**Soundness:** 3
**Presentation:** 3
**Contribution:** 2
**Rating:** 6
**Confidence:** 4

**Summary:**

The paper integrates SDMap information to complement limitations of on-board cameras for map construction. To enhance the ability of geometry prediction and topology reasoning, , they propose a topology-guided decoder. The proposed method achieves state-of-the-art results on OpenLaneV2, demonstrating that incorporating SDMap yields a significant improvement in accuracy.

**Strengths:**

1. The approach encodes geometry and road types from SDMap into features and integrates these into BEV features for use in the decoder, which improves performance.

2. To explore the mutual influence of topology and geometry, this work introduce a topology-guided self-attention mechanism to aggregate vicinity lane features.

**Weaknesses:**

1. *Performance Drop in Model Combination*: Combining LaneSegNet with P-MapNet results in decreased performance, which is unexpected and requires clarification. An explanation for this discrepancy, particularly given that P-MapNet also employs a cross-attention mechanism, would provide valuable insight into the interaction between the two models.

2. Limited Novelty in SDMap Encoding and Fusion: The methods used for map tokenization and fusion lack significant novelty, with SDMap encoding resembling SMERF’s approach and the fusion method similar to P-MapNet, both of which utilize cross-attention.

**Questions:**

1. Task Choice:

> Instead, researchers are focusing on online vectorized HDMap construction … However, sensor-only approaches still face challenges for long-range perception due to the limited field of view of camera, …

The paper’s abstract suggests a focus on addressing challenges in long-range perception due to camera field-of-view limitations. Given this:

  a. Why does this work emphasize the Topology task for incorporating SDMap rather than focusing on an HDMap task?

  b. For topology reasoning, why was the OpenLaneV2 **lane segment** task selected over the OpenLaneV2 **lane centerline** task?

2. Decoder Analysis

In the ablation study (Table 3, last two rows), the authors compare the performance impact of using the Topo-Guided Decoder based on an SDMap incorporation baseline. Have the authors considered evaluating the Topo-Guided Decoder on a baseline without SDMap integration? It would be helpful to understand whether this module maintains effectiveness in the absence of SDMap incorporation.

3. Generalizability of SDMap Fusion Method:

While the paper aims to leverage SDMap to address sensor limitations, it is unclear whether the proposed SDMap encoding and fusion method can also enhance performance in other BEV map-based tasks beyond the current setup.

Given that the combination of P-MapNet with LaneSegNet lowers LaneSegNet’s original performance (Table 1), additional experiments on other tasks would clarify the versatility and potential trade-offs of this fusion approach.

---

> ### Author Response · Authors · 2024-11-19
> **Author reponse to Reviewer  XgUo**
>
> We appreciate your valuable comments and questions. We hope our response can address your concerns.
>
> > ***Q1: Performance Drop in Model Combination for P-MapNet***
>
> A1: Thanks for your valuable suggestions. If I’m not mistaken, the "two models" you mentioned refer to LaneSegNet + SMERF (map tokenization) and LaneSegNet + P-MapNet. The decreased performance of the mAP may be attributed to several factors:
>
> **First**, P-MapNet uses cross-attention to fuse the 2D-grid SD feature and 2D-grid BEV feature, with a computational complexity of $O(H_{bev} \times W_{bev}  \times H_{SD}\times W_{SD})$. Because we use a high-resolution (200 x 100) setting following LaneSegNet, we must downsample their resolutions to mitigate computational overhead during cross-attention. Consequently, this downsampling inevitably sacrifices precision. Our reimplementation strictly follows the official code, which utilizes a CNN and a deconvolution network to downsample and recover the BEV size. In contrast, the cross-attention operation in our method and SMERF is computed between the SD tokens and BEV features with a complexity of $O(H_{bev} \times W_{bev}  \times N_{SD})$. Here $N_{SD} << H_{SD}\times W_{SD} $. Thus there is no need for downsampling the BEV size.
> **Second**, P-MapNet mainly validates its effectiveness on segmentation-based and polyline-based lane detection. However, there may be some differences between tasks when directly transferring their SD fusion design.
>
> > ***Q2: Limited Novelty in SDMap Encoding and Fusion***
>
> A2: We would like to reemphasize our contributions. While the map tokenization is similar to SMERF's, our spatial SDMap encoding and fusion differs significantly from P-MapNet.  We encode various attributes (e.g., road shape and curvature) into different channels of 2D grid maps (as illustrated in Figure 6). Moreover, in the spatial fusion process, we do not utilize a cross-attention mechanism; rather, we directly add the 2D spatial SD features to the BEV queries and BEV features, finally achieving unconflicted performance gains using the proposed SDMap spatial encoding and map tokenization.
>
> > ***Q3-a: Task Choice：
> > Why does this work emphasize the Topology task for incorporating SDMap rather than focusing on an HDMap task***
>
> A3-a: As HDMaps contain geometry and topology information of the map, we see HDMap construction as a general concept, which not only reconstructs the geometry of lanes but also predicts the topology.  Typical methods such as MapTR usually formulate this problem as a task of recognizing polylines of the map elements. Many works (e.g., TopoNet) have been proposed to advance HDmap reconstruction toward a more comprehensive and practical multi-task paradigm. As one of these, LaneSegNet solves the HDMap construction problem with a new representation of lane segments. We think it is a more comprehensive and challenging benchmark consisting of geometry prediction and topology reasoning, which is more applicable to real-world autonomous driving.
>
> Here we use the statement of "topology-enhanced" due to two aspects: (1) the SDMap information also contains the road topology information in a bird's eye view, which enhances the lane segmentation perception task; (2) we propose a topology-guided decoder to achieve mutual promotion between geometrical and topological features.
>
> > ***Q3-b: For topology reasoning, why was the OpenLaneV2 lane segment task selected over the OpenLaneV2 lane centerline task?***
>
> A3-b: The lane segment task was selected over the centerline task as it offers a more thorough geometric evaluation. In addition to topology assessment, the lane segment perception task evaluates both lane centerline and left/right boundary accuracy, whereas the centerline task is limited to centerline accuracy. This richer evaluation aligns with the emphasis on the accuracy of lane lines in previous research.
>
> > ***Q4: Decoder Analysis***
>
> A4: We appreciate your valuable suggestions. We will conduct the experiments as you recommended. Once we obtain the results, we will include them in the revision or address them during the rebuttal.
>
> > ***Q5: Generalizability of SDMap Fusion Method***
>
> A5: We thank you for your comments. As we point out above, one reason why we selected the lane segmentation task is that we suppose this benchmark contains many map-related tasks. We believe this benchmark provides a more comprehensive assessment of the overall performance of BEV mapping models. We may validate our method on other map tasks in the future.
>
> For the generalization of the SDMap fusion, we pay more attention to the versatility of the model when the input SDMaps have errors. In real applications, standard-definition (SD) maps provide road-level information that inevitably has meter-level errors. Our experiments show that a model tested with high precision under the accurate SDMap input performs worse when adding SDMap noise. For this, we give an in-depth analysis and some potential solutions.

---

> > ### Author Response · Authors · 2024-11-26
> > **Author response to Reviewer XgUo**
> >
> > Dear reviewer,
> >
> > To address the Q4 question, we conducted experiments under identical conditions to ensure a fair comparison. Specifically, we ran the official LaneSegNet code alongside the combination of LaneSegNet and the Topology Guided Decoder (TGD), excluding the influence of SDMap information. The results are as follows:
> >
> > | Method | mAP| AP$_{ls}$|AP$_{ped}$|TOP$_{lsls}$|
> > | ---| ---| ---|---|---|
> > | LaneSegNet| 31.8%|31.6%|32.1%|25.5%|
> > | LaneSegNet + TGD| 32.2% (+0.4) |30.7% (-0.9)|33.6% (+1.5)|28.2% (+2.7)|
> >
> > The results show a slight performance gain (0.4) on the mAP and an obvious gain on the topology metric, which is consistent with the performance gains shown in Table 3.
> >
> > We would sincerely appreciate it if we could get some feedback from you regarding the above concerns. Please also consider raising the score if all the raised issues are addressed.
> >
> > Best!
> > The authors of Paper #7200

---

> ### Comment · Reviewer_XgUo · 2024-11-26
>
> Thank you for addressing most of my concerns and questions in your response. Though, I still have some concerns regarding the response from the authors.
>
> **Q1-Response:**
> The comparison with P-MapNet appears unfair as you use a resolution of 200x100 for your method, while P-MapNet employs a lower resolution of 50x25 for both BEV-feats and SDMap-feats. This significant difference in resolution impacts the validity of the comparison. While I understand the efficiency considerations, you could provide a computational comparison to demonstrate your model’s advantages (as prior works suggest, higher-resolution features typically enhance perception performance.)
>
> Furthermore, you mentioned that
> > P-MapNet mainly validates its effectiveness on segmentation-based and polyline-based lane detection. However, there may be some differences between tasks when directly transferring their SD fusion design.
>
>  However, the employed LaneSegNet also appears to a polyline-based method. I think the performance decrease primarily results from the significantly lower resolution used.
>
> **Q2-Response:**
> Thank you for clarifying your spatial fusion process. However, based on the paper (L226-227), after summing the SD features and BEV features, a cross-attention mechanism is applied to further aggregate the summed features. I think this appears similar to the method employed by P-MapNet, which also utilizes cross-attention for (SD) feature integration.
>
> Again, I appreciate the efforts from authors in the response. While most concerns have been addressed, ensuring **fairness** in comparisons would further clarify the advantages of the proposed method (To clarify, this is a suggestion for future experiments, not a requirement for the authors to conduct the mentioned fair comparison here.)

---

> > ### Author Response · Authors · 2024-11-27
> > **Author response to Reviewer XgUo**
> >
> > We thank you for your detailed feedback and comments. We are happy to know that most concerns have been addressed.
> >
> > Regarding the comparison with P-MapNet, it is challenging to precisely replicate P-MapNet's approach for the lane segmentation task in OpenLaneV2. Thus establishing a truly fair comparison with P-MapNet is inherently challenging. In our understanding, the core of P-MapNet lies in its rasterized representation of the SDMap and the cross-attention fusion between the rasterized SD representation and the rasterized BEV features. We have faithfully adhered to both these fundamental principles in our implementation. We agree that the performance discrepancy may primarily stem from resolution differences, which is directly related to P-MapNet's cross-attention mechanism.
> >
> > The cross-attention described in Lines 226-227 involves BEV queries enhanced with SD features attending to image features (image features as keys/values) to aggregate visual information from surrounding camera views. This operator is inherently utilized in BEVFormer, which differs from the cross-attention operator in P-MapNet. In P-MapNet, the SDMap cross-attention computes pairwise interactions between 2D-grid BEV queries and 2D-grid SD features (SD features as keys/values), with the keys/values originating from different sources. And there is another cross-attention operator between BEV features and SD tokens, which is also distinct from the cross-attention in P-MapNet. Here, SD tokens, rather than 2D-grid SD features, are used as keys/values.
> >
> > We will consider your suggestions for future experiments to ensure fairness as much as possible. Thank you once again for your feedback.
> >
> > Best Regards,
> >
> > The authors of Paper #7200

---

> ### Comment · Reviewer_XgUo · 2024-11-27
>
> Thank you for the clarification, and I apologize for the confusion regarding the exact line numbers. In my original review comment (W2), I was referring to the second cross-attention mechanism, which uses SD tokens for cross-attention. This approach appears similar to P-MapNet while employing a tokenization method similar to that used in SMERF.  In my opinion, this approach is not truely innovative, but acceptable.
>
> Moreover, I noticed that in Table 5, you conducted the “LaneSegNet + P-MapNet” experiment with a BEV resolution of 100x50. However, in Table 1, the results for the same method are reported with a resolution of **50x25**, which is only **1/4** of the resolution used in your model experiment. Could you clarify the reason? Additionally, I am curious about the corresponding results of P-MapNet with a 100x50 resolution from Table 5. While still only 1/2 of your model’s resolution, it would provide a fairer comparison than 50x25.

---

> > ### Author Response · Authors · 2024-11-27
> > **Author response to Reviewer XgUo**
> >
> > Thanks for your feedback. We now understand your previous point.
> >
> > The first reason why we use a resolution of 50 x 25 is that we strictly follow the code of P-MapNet to downsample and upsample the BEV features using a CNN (https://github.com/jike5/P-MapNet/blob/b8b4cf2295ee75826046eef9cfa12b107fb43619/model/pmapnet_sd.py#L107)  and a deconvolution network (https://github.com/jike5/P-MapNet/blob/b8b4cf2295ee75826046eef9cfa12b107fb43619/model/pmapnet_sd.py#L118). The downsampling is to use a CNN with two convolutional layers with stride=2 for each layer so that the BEV resolution is downsampled from 200x100 to 50x25. Thus it can be seen as an alignment with the P-MapNet neural network architecture design.
> >
> > In addition, in Table 5, we aim to analyze the complexity, speed, and number of model parameters of different models. We tested the inference speed of the LaneSegNet + P-MapNet at the resolution of 100x50 but didn't train that model.  This is because it will compute cross-attention between two sequences with a length of 5000, leading to 5000×5000 attention operations. Such computational demands significantly increase GPU memory usage and slow down inference. While we believe this model would outperform its counterpart at a 50×25 resolution, the computational cost is prohibitively high. Notably, its FPS is only 3.3, making it slower than most models listed in Table 5.

---

> ### Comment · Reviewer_XgUo · 2024-11-27
>
> Thanks for the response. No more questions.

---

### Meta-Review · Area_Chair_gM7e · 2024-12-23

**Metareview:**

The paper proposes an approach to utilize on-board cameras combined with SDMap information to overcome the need for HDMaps in autonomous driving. A new topology-guided decoder is proposed to achieve state-of-the-art experimental results. However, the approach derives strongly from prior works like SMERF and P-MapNet, limiting its technical contribution. While the improvement in accuracy over prior works is noted, practical usage is not certain. Overall, based on the majority of reviewer opinions, the paper may not be accepted for ICLR. It is suggested for the authors to incorporate reviewer suggestions and resubmit to a future venue.

**Additional Comments On Reviewer Discussion:**

Drcu finds the contributions of the paper limited in comparison to prior works like UniHD Map and persists in the opinion following the author rebuttal. Clarifications sought by ckWD on SDMap details are provided by the rebuttal, but more comprehensive error analyses are not included, leading to a score leaning towards rejection. Similarly, numerous remaining errors preventing real-world usage are pointed out by sYjC who also recommends rejection. D5MY finds the contributions limited relative to prior works and while XgUo is the most positive reviewer, they share a similar concern. Overall, the reviews lean towards not accepting the paper and suggest numerous directions for improvement.

---

### Decision · Program_Chairs · 2025-01-22

Reject